# Weighted Geodesic Distance Following Fermat's Principle

**Facundo Sapienza**
Aristas SRL,
Buenos Aires, Argentina.
`f.sapienza@aristas.com.ar`

**Pablo Groisman**
IMAS-CONICET,
NYU-ECNU IMS at NYU Shanghai,
and Universidad de Buenos Aires, Argentina.
`pgroisma@dm.uba.ar`

**Matthieu Jonckheere**
Instituto de Cálculo-CONICET,
Universidad de Buenos Aires, Argentina.
`mjonckhe@dm.uba.ar`

## Abstract

We propose a density-based estimator for weighted geodesic distances suitable for data lying on a manifold of lower dimension than ambient space and sampled from a possibly nonuniform distribution. After discussing its properties and implementation, we evaluate its performance as a tool for clustering tasks. A discussion on the consistency of the estimator is also given.

## 1 Introduction

In most unsupervised learning tasks such as clustering, classification, recommendation, or dimensionality reduction, a notion of similarity between points is both crucial and usually not directly available as an input, Shaw et al. (2011). Instead, this distance has to be guessed or inferred from the data itself. This is the case when the data is in fact on an unknown manifold with lower dimension, which is a typical situation in most applications, Bengio et al. (2013). In these situations, the Euclidean distance is typically misleading; this effect, being more dramatic as dimension increases, Aggarwal et al. (2001).

Learning based on similarity has been considered in diverse applications: time series, Morse & Patel (2007), clustering of chemical structures, Barnard & Downs (1992), genetic data, Lawson & Falush (2012) and documents or texts, Wang et al. (2011).

Some historical linear methods as Principal Component Analysis can actually be thought as a first attempt to do this, Wall et al. (2003). More advanced powerful nonlinear procedures have also been developed to perform dimensionality reduction of the data by representing them in a lower dimensional space. Examples of such methods are Multidimensional scaling (MDS) Borg & Groenen (2003), t-distributed stochastic distance embedding (t-SNE) van der Maaten & Hinton (2008), Spectral embedding Belkin & Niyogi (2003) and Isometric mapping (Isomap) and $C$-Isomap Tenenbaum et al. (2000); de Silva & Tenenbaum (2002).

A particular feature of Isomap and $C$-Isomap is that they implement a distance estimator before performing the dimensionality reduction, building up a bridge between geodesic estimation and unsupervised learning. The seminal paper introducing Isomap, Tenenbaum et al. (2000) underlines the improved efficiency in dimensionality reduction and classification when considering adequate notions of distances. Particularly in the context of image analysis. However, Isomap is designed to deal with data with constant density and $C$-Isomap is designed to estimate the distance in the preimage for conformal embeddings, which could not be the case in many transformations. Neither of them takes into account the underlying density from which the points are sampled to define the geodesic estimator. In the sequel, we propose a new method to estimate distances in high dimensional non-uniform datasets that attacks this two issues.

## 2  $d$-DISTANCE ESTIMATOR

Let $\mathcal{M} \subseteq \mathbb{R}^P$ be a $D$-dimensional manifold. That is, $\mathcal{M}$ can be locally transformed into $\mathbb{R}^D$. Typically we have $D \ll P$, but this is not required. Consider a sample of $N$ points $\mathbb{X}_N \subset \mathcal{M}$. Let $\ell(\cdot, \cdot)$ be a distance defined on $\mathcal{M} \times \mathcal{M}$ (a typical choice could be Euclidean distance in $\mathbb{R}^P$, but other choices are allowed). For $d \geq 1$ and given two points $\mathbf{p}, \mathbf{q} \in \mathcal{M}$ we define the $d$-*distance estimator* as

$$\mathcal{D}_{\mathbb{X}_N}(\mathbf{p}, \mathbf{q}) = \min_{(\mathbf{x}_1, \ldots, \mathbf{x}_K) \in \mathbb{X}_N^K} \sum_{i=1}^{K-1} \ell(\mathbf{x}_i, \mathbf{x}_{i+1})^d. \tag{1}$$

The minimization is done over all $K \geq 2$ and all finite sequences of data points with $\mathbf{x}_1 = \operatorname{argmin}_{\mathbf{x} \in \mathbb{X}_N} \ell(\mathbf{x}, \mathbf{p})$ and $\mathbf{x}_K = \operatorname{argmin}_{\mathbf{x} \in \mathbb{X}_N} \ell(\mathbf{x}, \mathbf{q})$. Note that the curse of dimensionality effects, Aggarwal et al. (2001) are much stronger on the distance between those points that are faraway to each other than on those which are close to each other. With this in mind, for $d > 1$, the $d$-distance discourages consecutive points with large $\ell$. Also observe that $\mathcal{D}_{\mathbb{X}_N}$ verifies triangular inequality and so, it is indeed a distance. When $d = 1$, we recover the distance $\ell(\cdot, \cdot)$, but if $d > 1$, the $d$−distance tends to follow more closely the manifold structure and regions with high density values. We conjecture that for given $\mathbf{p}, \mathbf{q} \in \mathcal{M}$, if $\mathbb{X}_N$ is an i.i.d. sample with density $f \colon \mathcal{M} \to \mathbb{R}$, the estimator $\mathcal{D}_{\mathbb{X}_N}(\mathbf{p}, \mathbf{q})$ converges in the following sense:

$$\lim_{N \to \infty} N^\alpha \mathcal{D}_{\mathbb{X}_N}(\mathbf{p}, \mathbf{q}) = c_{d,D} \inf_{\Gamma \subset \mathcal{M}} \int_\Gamma \frac{1}{f^\alpha}. \tag{2}$$

Here $\alpha = \frac{d-1}{D}$ and $c_{d,D}$ is a constant that depends only on $d$ and $D$. The optimization is performed over all continuous rectifiable paths $\Gamma$ contained in the manifold that start at $\mathbf{p}$ and end at $\mathbf{q}$. In other words, we expect $N^\alpha \mathcal{D}_{\mathbb{X}_N}(\mathbf{p}, \mathbf{q})$ to be a consistent estimator of the right hand side of (2), which turns out to be a distance in $\mathcal{M}$ if $f$ is positive. Observe that this distance is a weighted geodesic distance in which paths are adjusted according to the value of $1/f^\alpha$.

We proved this fact in the simplest case in which $f$ is constant in a compact set. Since the strength of our estimator lies in the fact that it can deal with non-constant $f$, this result is not meaningful by itself. But we think it is a first step towards a full proof of (2). We can also observe that (2) holds in simulations. See supplementary material.

For $d > 1$, the $d$-distance is indeed taking into account the density $f$ imitating Fermat's principle for the light path[1], with $f^{-\alpha}$ playing the role of the refraction index.

### 2.1  ALGORITHM AND IMPLEMENTATION

In practice, the computation of the $d$-distance relies on local estimations. Although the optimization is defined on all possible paths using data points, we show in supplementary material that if $d > 1$, it can be restricted to local paths. Let $\mathcal{N}_k(\mathbf{x})$ be the set of $k$-nearest neighbors of $\mathbf{x}$ and let $\hat{\mathcal{D}}_{\mathbb{X}_N}(\mathbf{p}, \mathbf{q})$ be a new estimator defined as in (1) but with the additional constrain that $\mathbf{x}_{i+1} \in \mathcal{N}_k(\mathbf{x}_i)$ for each $i \leq K - 1$. Then it holds that given $\varepsilon > 0$ there is $k = \mathcal{O}(\log(N/\varepsilon))$ such that $\mathcal{D}_{\mathbb{X}_N}(\mathbf{p}, \mathbf{q}) = \hat{\mathcal{D}}_{\mathbb{X}_N}(\mathbf{p}, \mathbf{q})$ with probability $1 - \varepsilon$. In this way, the complexity can be reduced to $O(N^2 (\log N)^2)$ without modifying the asymptotic properties of the estimator.

## 3  CLUSTERING PROPERTIES

We compare the performance of the $d$-distance estimator for various values of $d$ and Isomap/$C$-Isomap for clustering tasks. We use the well-known example coined "Swiss roll", Figure 1(a) and 1(b). We consider a dataset composed of 4 subsets steaming from independent Normal distributions (restricted to the unit square) with mean $\mu_1 = (.3, .3), \mu_2 = (.3, .7), \mu_3 = (.7, .3), \mu_4 = (.7, .7)$ respectively and constant variance, Figure 1(a). Then, we apply the *Swiss Roll transformation*, Figure 1(b). Observe that the transformed data set is not Gaussian and each subset of transformed data has different variance. We then compute the $d$-distance between all pair of points and run (1000

---

[1] In optics, Fermat's principle states that the optical length of the path followed by light between two fixed points, $\mathbf{p}$ and $\mathbf{q}$, is a local minimum of the functional $\Gamma \mapsto \int_\Gamma \mathrm{n} \, dl$. Here n is the refractive index of the medium and the integral represents the time required by light to do the path $\Gamma$.

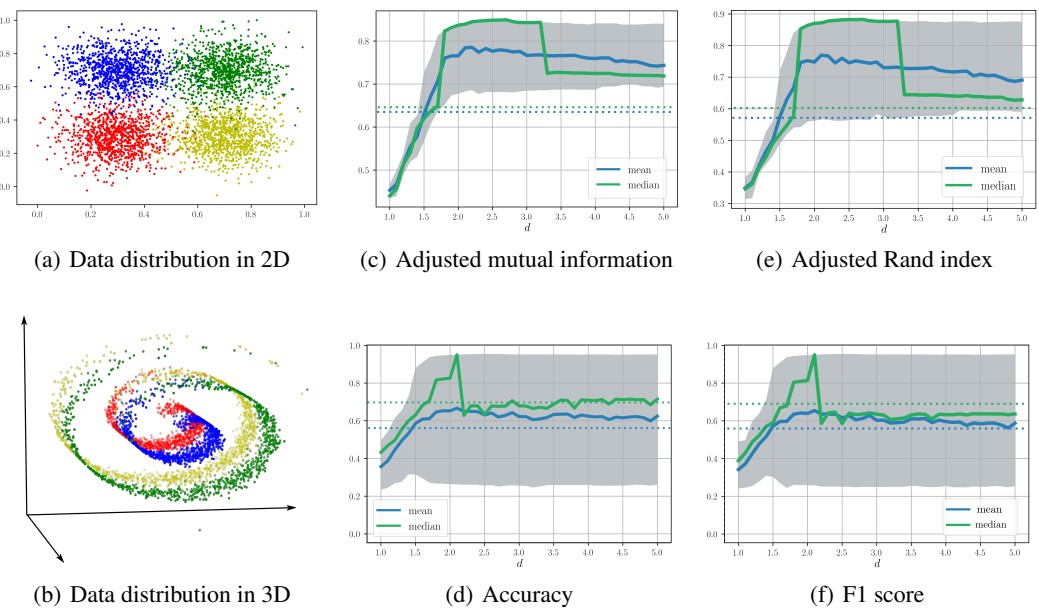

Figure 1: Clustering the non-uniform Swiss Roll data.

iterations with different initial conditions) the classical $K$-medoids clustering algorithm, Friedman et al. (2001), using this matrix of $d$-distances as input. To evaluate the clustering results, we compute the median (green line), the mean (blue line) and the inter-quantile range (gray) for the adjusted mutual information, Meilă (2007); Vinh et al. (2010), 1(c); the adjusted Rand index, Hubert & Arabie (1985), 1(e); the accuracy, 1(d), and the $F_1$ score Powers (2011), 1(f) for different values of $d$ and with $k = 300$. The case $d = 1$ corresponds to Euclidean distance and the dashed line corresponds to best output among Isomap and $C$-Isomap, for which the nearest-neighbors parameter was selected to maximize the performance of each index. For all performance criteria employed, there is a large range of the parameter $d$ where the $d$-distance behaves significantly better than Isomap. For the adjusted mutual information and adjusted Rand index we observe an optimal region for $1.8 \leq d \leq 3.2$, while for the accuracy and $F_1$ score this region is limited to $1.7 \leq d \leq 2.2$.

## 4  CONCLUSIONS

We proposed a new estimator for distances between points on an unknown manifold which takes into account the intrinsic density $f$ in order to bring closer two points if there is a (short) path between them lying on a region with high density values. This distance can be then used as an input for dimensionality reduction or clustering algorithms.

The choice of the parameter $d$ might depend on the specific application the $d$-distance is used for, but it has a clear meaning. It measures how we decide to weight the density values $f$. Observe that $d$ is not a parameter of the algorithm but of the macroscopic distance we want to estimate. Similar to the parameter $p$ in $p$-norms.

We evaluated its performance for clustering tasks in the classical Swiss Roll example with non-uniform distribution. In this case, we have shown that the use of the $d$-distance improves the clustering. In future work, we plan to study convergence properties of our estimator in more detail; to establish criteria to choose the value of $d$ and to use this criteria to perform dimensionality estimation in real data.

### ACKNOWLEDGMENTS

We thank Aristas SRL for the support with servers and programming. We would also like to thank Martín Arjovsky and Ezequiel Smucler for their suggestions that helped as to improve this work.

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

# Supplementary Material

## A  RINGS DATA SET

Let us illustrate further how the $d$-distance operates on the following simple example. Consider a set of points made of concentric rings of different radius with or without bridges, Figures 2(a) and 2(c). Consider first the data without bridges. The macroscopic $d$-distance (right hand side of (2)) between two points in different connected components is infinity, while two points in the same connected component have a finite distance. Computing the $d$-distance and representing this distance (for instance with t-SNE), we retrieve the connected sets, of course distorted in terms of Euclidean distance. Note that the distance relation between the rings is lost, as expected (Figure 2(b)). The fact that rings well separate in the absence of bridges sheds light on the $d$-distance leverages that can be used for clustering. When we consider the rings with bridges data we see that the $d$-distance can recover the geometric structure of the data as well (Figure 2(d)).

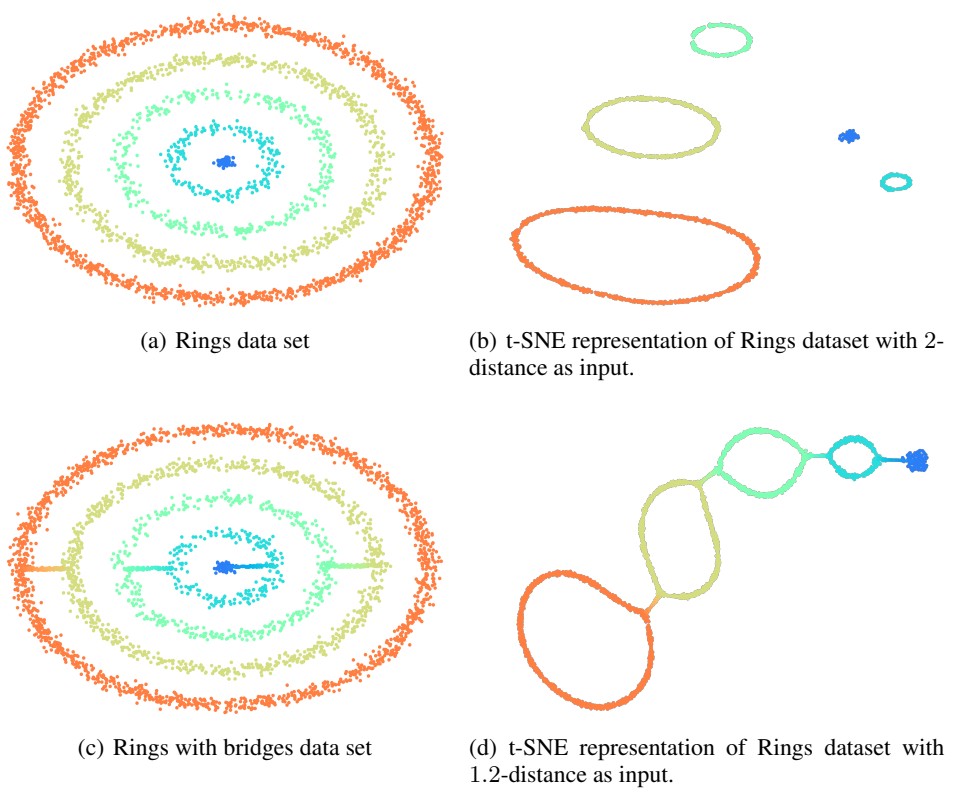

(a) Rings data set

(b) t-SNE representation of Rings dataset with 2-distance as input.

(c) Rings with bridges data set

(d) t-SNE representation of Rings dataset with 1.2-distance as input.

Figure 2: The $d$-distance enlarges the distance between points in different connected components. If we take into account the normalization factor $N^\alpha$, from (2) it can be seen that points separated from a region where the density $f$ is close to zero have large $d$-distance. While points that can be connected by a path lying in a region with high density, are close to each other.

## B  CONSISTENCY OF THE $d$-DISTANCE ESTIMATOR

In this section we prove the macroscopic limit of the $d$-distance estimator for the simple case in which $f$ is constant in a compact set $C \subset \mathbb{R}^D$.

### B.1 POISSON POINT PROCESSES

We start with the following results from Howard & Newman (1997). For a Borel set $A \subset \mathbb{R}^D$, denote $|A|$ its Lebesgue measure and $\#A$ the number of points in $A$. We also denote with $\|\cdot\|$ the Euclidean norm.

Let $\mathbb{X}$ be a locally finite subset of $\mathbb{R}^D$. We are going to be mainly interested in the case in which $\mathbb{X}$ is a Poisson Point Process in $\mathbb{R}^D$, Kallenberg (2002) but other cases are going to be treated as well. We refer to points in $\mathbb{X}$ as particles, to distinguish them from other points in $\mathbb{R}^D$. For any point $\mathbf{p} \in \mathbb{R}^D$, define the center of its Voronoi cell as

$$\mathbf{y}(\mathbf{p}) = \operatorname*{argmin}_{\mathbf{y}_i \in \mathbb{X}} \|\mathbf{p} - \mathbf{y}_i\|.$$

For any $\mathbf{p}, \mathbf{q} \in \mathbb{R}^D$, a finite sequence $\mathbf{y}_1, \ldots, \mathbf{y}_K$ of particles with $\mathbf{y}_1 = \mathbf{y}(\mathbf{p}), \mathbf{y}_K = \mathbf{y}(\mathbf{q})$ is called a *path* (or an $\mathbb{X}$-path if necessary) from $\mathbf{p}$ to $\mathbf{q}$. Given a parameter $d > 1$ we define the $d$-distance between $\mathbf{p}, \mathbf{q} \in \mathbb{R}^D$ with respect to $\mathbb{X}$ by

$$\mathcal{D}_{\mathbb{X}}(\mathbf{p}, \mathbf{q}) = \inf \left\{ \sum_{j=1}^{k} \|\mathbf{y}_{i+1} - \mathbf{y}_i\|^d \colon k \geq 2, \text{ and } (\mathbf{y}_1, \ldots, \mathbf{y}_k) \text{ is an } \mathbb{X}\text{-path from } \mathbf{p} \text{ to } \mathbf{q} \right\}.$$

Given a Borel set $C \subset \mathbb{R}^D$, a random locally finite configuration of points $\mathbb{X} \subset C$ is said to be a Poisson point process with intensity $\lambda > 0$ if for any pair of disjoint Borel sets $A, B \subset C$, we have

$$\mathbb{P}(\#(\mathbb{X} \cap A) = k, \#(\mathbb{X} \cap B) = j) = \frac{e^{-(\lambda|A|+\lambda|B|)}(\lambda|A| + \lambda|B|)^{k+j}}{k!\, j!}.$$

We observe that if $\mathbb{X}$ turns out to be a Poisson process, $\{\mathcal{D}_{\mathbb{X}}(\mathbf{p}, \mathbf{q})\}$ is a family of random variables indexed by $(\mathbf{p}, \mathbf{q}) \in \mathbb{R}^{2D}$ and almost surely, for any two points $\mathbf{p}, \mathbf{q}$ there is a unique path along which $\mathcal{D}_{\mathbb{X}}(\mathbf{p}, \mathbf{q})$ is realized. Let $\mathbf{0}$ be the origin of $\mathbb{R}^D$, then

**Proposition B.1** (Howard & Newman (1997), Lemma 3 and Lemma 4). *Let $\mathbb{X}$ be an intensity one Poisson process. There exists $0 < \mu < \infty$ such that*

$$\lim_{\|\mathbf{q}\| \to \infty} \frac{\mathcal{D}_{\mathbb{X}}(\mathbf{0}, \mathbf{q})}{\|\mathbf{q}\|} = \mu, \qquad almost\ surely. \tag{3}$$

### B.2 RANDOM NUMBER OF POINTS IN A COMPACT SET

We use a scaling argument to export the above result to our setting, in which we have an i.i.d sample in a compact set and the number of points goes to infinity. Let $C \subset \mathbb{R}^D$ be a convex compact set and let $\mathbb{X}_N = \{\mathbf{x}_1, \ldots, \mathbf{x}_{M_N}\}$ be a Poisson process in $C$ with intensity $N$. So that, $M_N$ is a Poisson random variable with parameter $|A|N$ and conditionally on $M_N = K$, $\mathbb{X}_N$ is a uniform i.i.d sample in $A$ of size $K$. Call $\lambda = 1/|C|$ the density.

**Proposition B.2.** *Let $\alpha = (d-1)/D$. For $\mathbf{p}, \mathbf{q}$ in the interior of $C$ we have*

$$\lim_{N \to \infty} N^{\alpha} \mathcal{D}_{\mathbb{X}_N}(\mathbf{p}, \mathbf{q}) = \mu \lambda^{-\alpha} \|\mathbf{p} - \mathbf{q}\|, \qquad in\ probability. \tag{4}$$

*Proof.* We first observe that, since the limit is deterministic, it is enough to prove convergence in distribution in (4). So, all the limits in this proof are in that sense. By translating, rotating and scaling $C$, we can assume without loss of generality that $\mathbf{p} = 0$ and $\mathbf{q} = \mathbf{e}_1 = (1, 0 \ldots, 0)$. For every $N$, the distribution of $\mathbb{X}_N$ coincides with the distribution of $\frac{1}{(\lambda N)^{1/D}} \mathbb{X} \cap C$. So (4) is equivalent to

$$\lim_{N \to \infty} \frac{\lambda^{\alpha} N^{\alpha}}{(\lambda N)^{d/D}} \mathcal{D}_{\mathbb{X} \cap (\lambda N)^{\frac{1}{D}} C}(0, (\lambda N)^{1/D} \mathbf{e}_1) = \mu.$$

The only difference between (4) and (3) is that in (3) the distance is minimized among $\mathbb{X}-$paths while in (4) the distance is minimized among $\mathbb{X} \cap (\lambda N)^{1/D} C$-paths. For any two points $\tilde{\mathbf{p}}$ and $\tilde{\mathbf{q}}$ and $a > 0$, consider the $a-$dilation of the segment from $\tilde{\mathbf{p}}$ to $\tilde{\mathbf{q}}$

$$[\![\tilde{\mathbf{p}}, \tilde{\mathbf{q}}]\!]_a := \{\mathbf{x} \colon \|\mathbf{x} - \mathbf{y}\| \leq a \text{ for some } \mathbf{y} \text{ in segment between } \tilde{\mathbf{p}} \text{ and } \tilde{\mathbf{q}}\}.$$

We will show the stronger result that, for every $a > 0$

$$\lim_{N \to \infty} \frac{\mathcal{D}_{\mathbb{X}}(\mathbf{0}, N\mathbf{e}_1)}{N} = \lim_{N \to \infty} \frac{\mathcal{D}_{\mathbb{X} \cap [\![\mathbf{0}, N\mathbf{e}_1]\!]_{aN}}(\mathbf{0}, N\mathbf{e}_1)}{N}. \tag{5}$$

Assume there is an infinite sequence $(N_k)_{k \geq 1}$ such that the minimizing path in $\mathcal{D}_{\mathbb{X}}(\mathbf{0}, N_k\mathbf{e}_1)$ contains at least one particle $\mathbf{q}_k$ in the complement of $[\![\mathbf{0}, N\mathbf{e}_1]\!]_{aN}$. For a such values of $k$ we have

$$\frac{\mathcal{D}_{\mathbb{X}}(\mathbf{0}, N_k\mathbf{e}_1)}{N_k} = \frac{\mathcal{D}_{\mathbb{X}}(\mathbf{0}, \mathbf{q}_k)}{N_k} + \frac{\mathcal{D}_{\mathbb{X}}(\mathbf{q}_k, N_k\mathbf{e}_1)}{N_k}.$$

By Proposition B.1, as $k \to \infty$ we have

$$\liminf_{k \to \infty} \frac{\mathcal{D}_{\mathbb{X}}(\mathbf{0}, N_k\mathbf{e}_1)}{N_k} \geq \mu \liminf_{k \to \infty} \frac{\|\mathbf{q}_k\|}{N_k} + \frac{\|\mathbf{q}_k - N_k\mathbf{e}_1\|}{N_k} \geq \mu\sqrt{1 + 2a} > \mu.$$

A contradiction with (3) that proves (5) and hence the proposition.

$\square$

## B.3 $N$ POINTS IN A COMPACT SET

First observe that given two locally finite configurations $\mathbb{X}$ and $\hat{\mathbb{X}}$, if $\mathbb{X} \subseteq \hat{\mathbb{X}}$, then

$$\mathcal{D}_{\mathbb{X}}(\mathbf{p}, \mathbf{q}) \leq \mathcal{D}_{\hat{\mathbb{X}}}(\mathbf{p}, \mathbf{q}).$$

**Lemma B.1** (Coupling, Kallenberg (2002)). *Let $\mathbb{X}$ and $\hat{\mathbb{X}}$ be Poisson point processes with intensities $0 < \lambda_1 < \lambda_2$ respectively. Then both processes can be constructed in the same probability space $(\Omega, \mathcal{F}, \mathbb{P})$ in such a way that with probability one $\mathbb{X} \subseteq \hat{\mathbb{X}}$.*

**Proposition B.3.** *Let $C \subset \mathbb{R}^D$ be a compact set and $\mathbb{X}_n = \{\mathbf{x}_1, \ldots, \mathbf{x}_N\}$ i.i.d random variables with uniform distribution in $C$. Call $\lambda = 1/|C|$. Then*

$$\lim_{N \to \infty} N^\alpha \mathcal{D}_{\mathbb{X}_N}(\mathbf{p}, \mathbf{q}) = \frac{\mu}{\lambda^\alpha} \|\mathbf{p} - \mathbf{q}\|.$$

*Proof.* For $\varepsilon > 0$, let $\mathbb{X}_N^+$, $\mathbb{X}_N^-$ be Poisson point processes in $C$ with intensities $N(1+\varepsilon)$, $N(1-\varepsilon)$ respectively and denote $M_N^+ = \#(\mathbb{X}_N^+ \cap C)$, $M_N^- = \#(\mathbb{X}_N^- \cap C)$. Then,

$$\lim_{N \to \infty} \frac{M_N^+}{N} = 1 + \varepsilon, \qquad \lim_{N \to \infty} \frac{M_N^-}{N} = 1 - \varepsilon, \qquad \text{a.s.}$$

In particular, with probability one, for $N$ large enough,

$$M_N^- N \leq \left(1 - \frac{\varepsilon}{2}\right) N < \left(1 + \frac{\varepsilon}{2}\right) N \leq M_N^+ N.$$

Due to Lemma B.1, we can construct $\mathbb{X}_N^+, \mathbb{X}_N^-$ and $\mathbb{X}_N$ in the same probability space in such a way that in the event $\Omega_N = \{M_N^- \leq N \leq M_N^+\}$ we have $\mathbb{X}_N^- \subseteq \mathbb{X}_N \subseteq \mathbb{X}_N^+$. Since $\mathbb{P}(\Omega_N) \to 1$, by Proposition B.2 we have for $\omega \in \Omega_N$,

$$N^\alpha \mathcal{D}_{\mathbb{X}_N^+}(\mathbf{p}, \mathbf{q}) \leq N^\alpha \mathcal{D}_{\mathbb{X}_N}(\mathbf{p}, \mathbf{q}) \leq N^\alpha \mathcal{D}_{\mathbb{X}_N^-}(\mathbf{p}, \mathbf{q}).$$

But the left hand side converges to $\frac{\mu\lambda^{-\alpha}}{1+\varepsilon}\|\mathbf{p} - \mathbf{q}\|$ and the right hand side converges to $\frac{\mu\lambda^{-\alpha}}{1-\varepsilon}\|\mathbf{p} - \mathbf{q}\|$. This implies that for every $\varepsilon > 0$

$$\frac{\mu\lambda^{-\alpha}}{1+\varepsilon}\|\mathbf{p} - \mathbf{q}\| \leq \liminf_{N \to \infty} N^\alpha \mathcal{D}_{\mathbb{X}_N}(\mathbf{p}, \mathbf{q}) \leq \limsup_{N \to \infty} N^\alpha \mathcal{D}_{\mathbb{X}_N}(\mathbf{p}, \mathbf{q}) \leq \frac{\mu\lambda^{-\alpha}}{1-\varepsilon}\|\mathbf{p} - \mathbf{q}\|.$$

We conclude

$$\lim_{N \to \infty} N^\alpha \mathcal{D}_{\mathbb{X}_N}(\mathbf{p}, \mathbf{q}) = \mu\lambda^{-\alpha}\|\mathbf{p} - \mathbf{q}\|.$$

To be more precise, for $\delta > 0$, call $N^\alpha \mathcal{D}_{\mathbb{X}_N}(\mathbf{p}, \mathbf{q}) =: \xi_N$ and $L := \mu\lambda^{-\alpha}\|\mathbf{p} - \mathbf{q}\|$. Then

$$\mathbb{P}(|\xi_N - L| > \delta) \leq \mathbb{P}(|\xi_N - L| > \delta | \Omega_N)\mathbb{P}(\Omega_N) + \mathbb{P}(\Omega_N^c) \to 0, \qquad N \to \infty.$$

$\square$

## C  Behavior of the $d$-distance for non-constant densities: Fermat's principle

In this section we present synthetic data as additional evidence for the asymptotic behavior of our estimator as $N \to \infty$. Figure 3 shows different optimal paths for the $d$-distance for different values of $d$. Points are sampled from a bi-dimensional distribution with density given by $f(\mathbf{x}) = \lambda_1 \mathbb{1}_{A_1}(\mathbf{x}) + \lambda_2 \mathbb{1}_{A_2}(\mathbf{x})$, with $\lambda_1 > \lambda_2 > 0$. In this case, Fermat's principle is equivalent to Snell's law: when the path crosses from one region to the other, since they have different densities, the path breaks (black line).

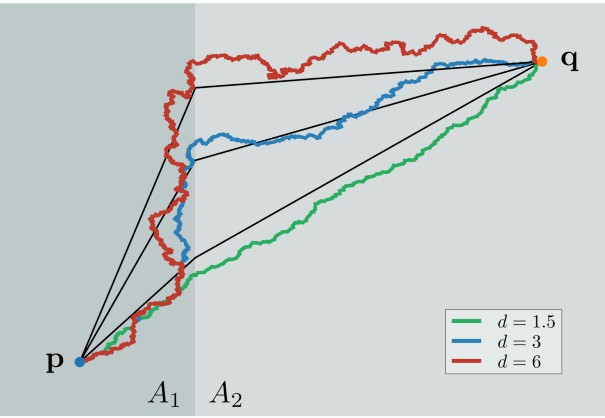

Figure 3: Optimal paths for the $d$-distance. The black line corresponds to the theoretical calculation of the macroscopic optimal path (equation (2)).

## D  Restriction to $k$-nearest neighbors

In this section we prove that the $d$-distance estimator does not change significantly if the minimization is performed over paths in wich consecutive particles are choosen among the $k$-nearest neighbors. The parameter $k$ depends on $N$ and the precision $\varepsilon$.

Let $\mathcal{N}_k(\mathbf{x})$ be the set of $k$-nearest neighbors of $\mathbf{x}$ and define

$$\hat{\mathcal{D}}_{\mathbb{X}_N}(\mathbf{p}, \mathbf{q}) = \min_{\substack{(\mathbf{x}_1, \ldots, \mathbf{x}_K) \in \mathbb{X}_N^K, \\ , \mathbf{x}_1 = \arg\min_{\mathbf{x} \in \mathbb{X}_N} \ell(\mathbf{x}, \mathbf{p}) \\ \mathbf{x}_K = \arg\min_{\mathbf{x} \in \mathbb{X}_N} \ell(\mathbf{x}, \mathbf{q}) \\ \mathbf{x}_{i+1} \in \mathcal{N}_k(\mathbf{x}_i)}} \sum_{i=1}^{K-1} \ell(\mathbf{x}_{i+1}, \mathbf{x}_i)^d. \tag{6}$$

**Proposition D.1.** *For every $\varepsilon > 0$ and $k = \mathcal{O}(\log(N/\varepsilon))$, with probability $1 - \varepsilon$ we have that $\hat{\mathcal{D}}_{\mathbb{X}_N}(\mathbf{p}, \mathbf{q}) = \mathcal{D}_{\mathbb{X}_N}(\mathbf{p}, \mathbf{q})$. More precisely, the minimizing path $\mathbf{y}_1^*, \ldots, \mathbf{y}_K^*$ verifies $\mathbf{y}_{i+1}^* \in \mathcal{N}_k(\mathbf{y}_i^*)$ for all $i = 1, \ldots, K - 1$ with probability $1 - \varepsilon$.*

The following lemma gives a upper bound for the probability that two consecutive particles in the path are $k$-nearest neighbors. Proposition D.1 follows easily from that. Denote $B(\mathbf{x}, r) = \{\mathbf{y} : \|\mathbf{y} - \mathbf{x}\| < r\}$ the open ball centered at $\mathbf{x}$ with radius $r$ and $\omega_D = |B(\mathbf{0}, 1)|$.

**Lemma D.1.** *Let $k_0 \in \mathbb{N}$. Let $K_N$ be the length of the optimal, path and for each $i \in \mathbb{N}$, let $F_i = \{K_N \geq i\}$ be the event that the optimal path has at least $i$ particles. Then, there exists $c_1, \delta > 0$ such that,*

$$\mathbb{P}\left(\mathbf{y}_{i+1}^* \notin N_{k_0}(\mathbf{y}_i^*), F_i\right) \leq \frac{c_1}{\delta^2} (1 - \delta)^{k_0 + 1}, \tag{7}$$

*Proof of Lemma D.1.* For $k \in \mathbb{N}$, denote with $k(\mathbf{y}_i^*)$ the $k$-nearest neighbor of $\mathbf{y}_i^*$ and consider the event

$$E_i^k = \left\{ \mathbf{y}_{i+1}^* = k(\mathbf{y}_i^*) \right\} \cap F_i.$$

Since there is no particle in the optimal path between $\mathbf{y}_i^*$ and $\mathbf{y}_{i+1}^*$, in the region defined by

$$B_i = \left\{ \mathbf{x} : \|\mathbf{y}_i^* - \mathbf{x}\|^d + \|\mathbf{y}_{i+1}^* - \mathbf{x}\|^d < \|\mathbf{y}_{i+1}^* - \mathbf{y}_i^*\|^d \right\},$$

we have $B_i \cap \mathbb{X}_N = \emptyset$. Define $r_i = \|\mathbf{y}_{i+1}^* - \mathbf{y}_i^*\|$. It is easy to see that $B_i \cap B(\mathbf{y}_i^*, r_i)$ has non-empty interior and that there is a deterministic constant $\delta > 0$, such that $B_{\delta r_i}(\mathbf{x}) \subset B_i$ for some $\mathbf{x} \in B_i$. Furthermore, $\{\mathbf{y}_{i+1}^* = k(\mathbf{y}_i^*)\} = \{\#(B(\mathbf{y}_i^*, r_i) \cap \mathbb{X}_N) = k - 1\}$. Then

$$E_i^k \subset$$

$$\left\{ i \leq K_N, |B(\mathbf{y}_i^*, r_i) \cap \mathbb{X}_N| = k - 1, \right.$$
$$\text{there is } \mathbf{x} \in B_i : B(\mathbf{x}, \delta r_i) \subset B(\mathbf{y}_i^*, r_i),$$
$$\left. \#(B(\mathbf{x}, \delta r_i) \cap \mathbb{X}_N) = 0 \right\}$$

$$\subset$$

$$\left\{ \text{there is } r > 0 : \#(B(\mathbf{y}_i^*, r) \cap \mathbb{X}_N) = k - 1, \right.$$
$$\text{there is } \mathbf{x} \text{ with } B(\mathbf{x}, \delta r) \subset B(\mathbf{y}_i^*, r),$$
$$\left. \#(B(\mathbf{x}, \delta r) \cap \mathbb{X}_N) = 0, \, i \leq K_N \right\}.$$

Let us consider a family $\{C_i\}_{i \leq N_{\text{tot}}}$ where each $C_i$ is a open $D$-hypercube with length $\delta r / \sqrt{D}$ that satisfies

$$C \subset \bigcup_{i=1}^{N_{\text{tot}}} \overline{C_i} \quad , \quad C_i \cap C_j = \emptyset \quad i \neq j$$

Then, it is clear that any ball of radius $r$ has non empty intersection with as much

$$N_{\text{cub}} = \frac{\omega_D (2r)^D}{(\delta r)^D D^{-D/2}} = \frac{2^D \omega_D D^{D/2}}{\delta^D}$$

cubes and that any ball of radius $\delta r$ contains at least one cube. Then there is $p > 0$, which depends linearly only on $\delta^D$ such that, regardless the value of $r$, we have $\mathbb{P}(E_i^k) \leq N_{\text{cub}}(1 - p)^k$. Then,

$$\mathbb{P}\left( \mathbf{y}_{i+1}^* \notin N_{k_0}(\mathbf{y}_i^*), F_i \right) = \sum_{k=k_0+1}^{\infty} \mathbb{P}(E_i^k) \leq \frac{c_1}{\delta^2} (1 - \delta)^{k_0+1}$$

$\square$

Notice that the events $\{\mathbf{y}_{i+1}^* \in \mathcal{N}_{k_0}(\mathbf{y}_i^*)\}$ and $\{\mathbf{y}_{i+2}^* \in \mathcal{N}_{k_0}(\mathbf{y}_{i+1}^*)\}$ are not independent. However, it is possible to bound the event that all consecutive particles in the optimal path are $k$-nearest neighbors, for $Nok$ large enough.

*Proof of Proposition D.1.* For fixed $k_0$, the probability that for all pairs of consecutive particles are $k_0$-nearest neighbors is

$$\mathbb{P}\left( \bigcap_{i=1}^{K-1} \{\mathbf{y}_{i+1}^* \in \mathcal{N}_{k_0}(\mathbf{y}_i^*)\} \right) = 1 - \mathbb{P}\left( \bigcup_{i=1}^{K-1} \{\mathbf{y}_{i+1}^* \notin \mathcal{N}_{k_0}(\mathbf{y}_i^*)\} \cap F_i \right)$$

$$\geq 1 - \sum_{i=1}^{K-1} \mathbb{P}\left( \mathbf{y}_{i+1}^* \notin \mathcal{N}_{k_0}(\mathbf{y}_i^*), F_i \right)$$

$$\geq 1 - \frac{N c_1}{\delta^2} (1 - \delta)^{k_0+1}$$

$$\geq 1 - \varepsilon,$$

if $k_0$ is chosen in order to satisfy $\frac{N c_1}{\delta^2} (1 - \delta)^{k_0+1} \leq \varepsilon$. That is, $k_0 = \mathcal{O}(\log(N/\varepsilon))$. $\square$

