# OpenReview forum: "Weighted Geodesic Distance Following Fermat's Principle"
_ICLR.cc/2018/Workshop — Accept_

### Official Review · AnonReviewer2 · 2018-03-09
**An interesting paper with strong theoretical analysis, but somewhat lacking in the explanation of its implementation**

**Rating:** 8
**Confidence:** 3

**Review:**

This paper defines a d-distance and its estimator and shows that it can be used for clustering. The definition of the distance depends on a parameter d. The theoretical analysis in equation (2) implies that the distance takes the density of the sampling into consideration. That is, the distance between two points are larger if the density between them is lower.

The idea in this paper is elegant and straightforward, the theoretical analysis seems convincing, and the simulation results look promising.

The disadvantage is about its implementation. That is, how is the proposed distance calculated in practice? The minimization problem (1) looks very difficult, and the current explanation seems unclear to me. From proposition D.1 and (6), it seems that there are k choices for each point x_i, so a greedy search would have a complexity of k^K=O(\log^K N), instead of O(\log^2 N) as in Section 2.1 (they are equivalent unless K=2, but I assume that K should be a large number)?

---

> ### Author Response · Authors · 2018-04-22
> **Response to AnonReviewer2**
>
> Thank you very much for your feedback!
>
> On the implementation: the d-distance between one point and the each of the remaining N-1 point is computed using Dijkstra's algorithm with a Fibonacci heap. Given a graph (E,V), the running time of this implementation is reduced to O(|E| + |V | log |V |). If we consider the k-nearest neighbors graph with k=O(log N), as Proposition D1 states, then |V|=O(N logN) and the running time to compute the d-distance between a given point p and the remaining ones is O(N log^2 N). Consequently, computing the d-distance between every two points in the data set is O(N^2 log^2 N).

---

### Official Review · AnonReviewer3 · 2018-03-14
**Interesting proposed estimator**

**Rating:** 7
**Confidence:** 2

**Review:**

The paper proposes a density-based estimator for weighted geodesic distances on low dimensional manifolds. Given a set of sampled points from the manifold and two given points p and q, the estimator for the weighted geodesic distance between p and q is the shortest path connecting p and q using points from the sample (the cost for each edge can be raised to some power). The authors conjectured that it's a consistent estimator for the weighted geodesic distance where the weight is related to the density. They proved this for the simple case where the density is constant over a compact set, and provided simulations supporting the conjecture.

Overall it's an interesting estimator, though more studies from theoretical and empirical sides are needed.

---

> ### Author Response · Authors · 2018-04-22
> **Response to AnonReviewer3**
>
> Thank you for your comments! Yes, we are actually working on a complete proof of the consistency of our estimator and in applications with real data.

---

### Official Review · AnonReviewer1 · 2018-03-15
**A new (unclear) metric**

**Rating:** 5
**Confidence:** 2

**Review:**

The authors propose a new metric for manifold learning, with some similarities to both the geodesic distance and the Minkowski norm (through the exponent d).

My main comment is that the paper is not clear enough to assess its merit.
Section 2, which is key in presenting the new metric, seems incomplete to me.
Triangular equality should be explicitly proven and equ. 2 looks obscure and should thus be much more carefully explained.
There is no integration variable in the integral (dx?).

How does the prposed distance relate to the geodesic distance?
The underlying question here is that a graph approximation of the geodesic distance works in similar way, since non neighbours have a distance that is supposed to be infinite, whereas in the proposed distance those points would be penalised similarly with the exponent d.
And in the end the authors appear to work with local neighbourhoods anyway, probably for to keep computation fast and tractable.

In the experiments, the respective statuses of Isomap and C-isomap is not clear. Usually these two methods are not considered interchangeable, as far as I know.
The clustering application somehow calls for a comparison with commute time distances and similar concepts, which are typically used (implicitly) in Laplacian eigenmaps and spectral clustering.

All in all, the paper is extremely difficult to assess.
While the proposed idea could be interesting, there are key questions that arise and that are not answered in this short format. The suppl.mat. dives into mathematical details without giving any practical clue.
The focus and structure of the paper could be largely improved, IMHO.

---

> ### Author Response · Authors · 2018-04-22
> **Response to AnonReviewer1**
>
> Thanks for the comments! We aim at developing these ideas in a longer format soon.
> Some further answers:
>
> - The notation with no integration variable is standard when there is no need to specify it.
> - We think that the reader can prove triangular inequality with no difficulty. Due to lack of space, we could not display all our computations.
> - The distance is macroscopically a weighted version of the geodesic distance as explained by our main result. In particular, it depends on the underlying density.
> - Though it can also be implemented using a graph of k_N nearest neighbors, our distance differs fundamentally from Isomap or C-Isomap as it takes into account the underlying density through the given d-norm penalization.
> - We did not claim that Isomap and C-isomap were interchangeable. We just compared our distance with their (best) performance.
> - We plan to conduct thorough experiments on clustering applications to fully illustrate the advantages of this distance.

---

### Comment · AnonReviewer1 · 2018-03-09
**Citation problem**

There is an issue with the citation format in the text.

---

### Decision · Program_Chairs · 2018-03-20
**ICLR 2018 Workshop Acceptance Decision**

**Decision:**

Accept

**Comment:**

Congratulations, your paper was accepted to the ICLR workshop.